# Speeding up Permutation Testing in Neuroimaging *

**Chris Hinrichs**†    **Vamsi K. Ithapu**†    **Qinyuan Sun**†    **Sterling C. Johnson**§†    **Vikas Singh**†

§William S. Middleton Memorial VA Hospital    †University of Wisconsin–Madison
`{hinrichs,vamsi}@cs.wisc.edu`   `{qsun28}@wisc.edu`
`{scj}@medicine.wisc.edu`   `{vsingh}@biostat.wisc.edu`
`http://pages.cs.wisc.edu/˜vamsi/pt_fast`

## Abstract

Multiple hypothesis testing is a significant problem in nearly all neuroimaging studies. In order to correct for this phenomena, we require a reliable estimate of the Family-Wise Error Rate (FWER). The well known Bonferroni correction method, while simple to implement, is quite conservative, and can substantially under-power a study because it ignores dependencies between test statistics. Permutation testing, on the other hand, is an exact, non-parametric method of estimating the FWER for a given $\alpha$-threshold, but for acceptably low thresholds the computational burden can be prohibitive. In this paper, we show that permutation testing in fact amounts to populating the columns of a very large matrix $\mathbf{P}$. By analyzing the spectrum of this matrix, under certain conditions, we see that $\mathbf{P}$ has a low-rank plus a low-variance residual decomposition which makes it suitable for highly sub–sampled — on the order of 0.5% — matrix completion methods. Based on this observation, we propose a novel permutation testing methodology which offers a large speedup, without sacrificing the fidelity of the estimated FWER. Our evaluations on four different neuroimaging datasets show that a computational speedup factor of roughly $50\times$ can be achieved while recovering the FWER distribution up to very high accuracy. Further, we show that the estimated $\alpha$-threshold is also recovered faithfully, and is stable.

## 1 Introduction

Suppose we have completed a placebo-controlled clinical trial of a promising new drug for a neurodegenerative disorder such as Alzheimer's disease (AD) on a small sized cohort. The study is designed such that in addition to assessing improvements in standard cognitive outcomes (e.g., MMSE), the purported treatment effects will also be assessed using Neuroimaging data. The rationale here is that, even if the drug does induce variations in cognitive symptoms, the brain changes are observable *much earlier* in the imaging data. On the imaging front, this analysis checks for statistically significant differences between brain images of subjects assigned to the two trial arms: treatment and placebo. Alternatively, consider a second scenario where we have completed a neuroimaging research study of a particular controlled factor, such as genotype, and the interest is to evaluate *group-wise* differences in the brain images: to identify which regions are affected as a function of class membership. In either cases, the standard image processing workflow yields for each subject a 3-D image (or voxel-wise "map"). Depending on the image modality acquired, these maps are of cerebral gray matter density, longitudinal deformation (local growth or contraction) or metabolism. It is assumed that these maps have been 'co-registered' across different subjects so that each voxel corresponds to approximately the same anatomical location. [1, 2].

In order to *localize* the effect under investigation (i.e., treatment or genotype), we then have to calculate a very large number (say, $v$) of univariate voxel-wise statistics – typically up to several million voxels. For example, consider group-contrast $t$-statistics (here we will mainly consider $t$-statistics, however other test statistics are also applicable, such as the $F$ statistic used in ANOVA testing, Pearson's correlation as used in functional imaging studies, or the $\chi^2$ test of dependence between variates, so long as certain conditions described in Section 2.3 are satisfied). In some voxels, it may turn out that a group-level effect has been indicated, but it is not clear right away what its true significance level should be, if any. As one might expect, given the number of hypotheses tests $v$, multiple testing issues in this setting are quite severe, making it difficult to assess the true Family-Wise Type I Error Rate (FWER) [3]. If we were to address this issue via Bonferroni correction [4], the enormous number of separate tests implies that certain weaker signals will almost certainly never be detected, even if they are real. This directly affects studies of neurodegenerative disorders in which atrophy proceeds at a very slow rate and the therapeutic effects of a drug is likely to be mild to moderate anyway. This is a critical bottleneck which makes localizing real, albeit slight, short-term treatment effects problematic. Already, this restriction will prevent us from using a smaller sized study (fewer subjects), increasing the cost of pharmaceutical research. In the worst case, an otherwise real treatment effect of a drug may not survive correction, and the trial may be deemed a failure.

**Bonferroni versus true FWER threshold.** Observe that theoretically, there *is* a case in which the Bonferroni corrected threshold is close to the true FWER threshold: when point-wise statistics are i.i.d. If so, then the extremely low Bonferroni corrected $\alpha$-threshold crossings effectively become mutually exclusive, which makes the Union Bound (on which Bonferroni correction is based) nearly tight. However, when variables are highly *dependent* – and indeed even without smoothing there are many sources of strong non-Gaussian dependencies between voxels, the true FWER threshold can be much more relaxed, and it is precisely this phenomenon which drives the search for alternatives to Bonferroni correction. Thus, many methods have been developed to more accurately and efficiently estimate or approximate the FWER [5, 6, 7, 8], which is a subject of much interest in statistics [9], machine learning [10], bioinformatics [11], and neuroimaging [12].

**Permutation testing.** A commonly used method of directly and non-parametrically estimating the FWER is Permutation testing [12, 13], which is a method of sampling from the Global (i.e., Family-Wise) Null distribution. Permutation testing ensures that any relevant dependencies present in the data carry through to the test statistics, giving an unbiased estimator of the FWER. If we want to choose a threshold sufficient to exclude *all* spurious results with probability $1 - \alpha$, we can construct a histogram of sample maxima taken from permutation samples, and choose a threshold giving the $1 - \alpha/2$ quantile. Unfortunately, reliable FWER estimates derived via permutation testing come at excessive (and often infeasible) computational cost – often tens of thousands or even millions of permutation samples are required, each of which requires a complete pass over the entire data set. This step alone can run from a few days up to many weeks and even longer [14, 15].

Observe that the very same dependencies between voxels, that forced the usage of permutation testing, indicate that the overwhelming majority of work in computing so many highly correlated Null statistics is redundant. Note that regardless of their description, strong dependencies of almost any kind will tend to concentrate most of their co-variation into a low-rank subspace, leaving a high-rank, low-variance residual [5]. In fact, for Genome wide Association studies (GWAS), many strategies calculate the 'effective number' ($M_{\text{eff}}$) of independent tests corresponding to the rank of this subspace [16, 5]. This paper is based on the observation that such a low-rank structure must also appear in permutation test samples. Using ideas from online low-rank matrix completion [17] we can sample a few of the Null statistics and reconstruct the remainder as long as we properly account for the residual. This allows us to sub-sample at *extremely low rates*, generally $< 1\%$. The **contribution** of our work is to significantly speed up permutation testing in neuroimaging, delivering running time improvements of up to $50\times$. In other words, our algorithm does the same job as permutation testing, but takes anywhere from a few minutes up to a few hours, rather than days or weeks. Further, based on recent work in random matrix theory, we provide an analysis which sheds additional light on the use of matrix completion methods in this context. To ensure that our conclusions are not an artifact of a specific dataset, we present strong empirical evidence via evaluations on four separate neuroimaging datasets of Alzheimer's disease (AD) and Mild Cognitive Impairment (MCI) patients as well as cognitively healthy age-matched controls (CN), showing that the proposed method can recover highly faithful Global Null distributions, while offering substantial speedups.

## 2 The Proposed Algorithm

We first cover some basic concepts underlying permutation testing and low rank matrix completion in more detail, before presenting our algorithm and the associated analysis.

### 2.1 Permutation testing

Randomly sampled permutation testing [18] is a methodology for drawing samples under the Global (Family-Wise) Null hypothesis. Recall that although point-wise test statistics have well characterized univariate Null distributions, the sample maximum usually has no analytic form due to the strong correlations across voxels. Permutation is particularly desirable in this setting because it is free of any distribution assumption whatsoever [12]. The basic idea of permutation testing is very simple, yet extremely powerful. Suppose we have a set of labeled high dimensional data points, and a univariate test statistic which measures some interaction between labeled groups for every dimension (or feature). If we randomly permute the labels and recalculate each test statistic, then by construction we get a sample from the Global Null distribution. The maximum over all of these statistics for every permutation sample is then used to construct a histogram, which therefore is a non-parametric estimate of the distribution of the sample maximum of Null statistics. For a test statistic derived from the real labels, the FWER corrected $p$-value is then equal to the fraction of permutation samples which were *more extreme*. Note that all of the permutation samples can be assembled into a matrix $\mathbf{P} \in \mathbb{R}^{v \times T}$ where $v$ is the number of comparisons (voxels for images), and $T$ is the number of permutation samples.

There is a drawback to this approach, however. Observe that it is in the nature of random sampling methods that we get many samples from near the mode(s) of the distribution of interest, but fewer from the tails. Hence, to characterize the threshold for a small portion of the tail of this distribution, we must draw a very large number of samples just so that the estimate converges. Thus, if we want an $\alpha = 0.01$ threshold from the Null sample maximum distribution, we require many thousands of permutation samples — each requires randomizing the labels and recalculating all test statistics, a very computationally expensive procedure when $v$ is large. To be certain, we would like to ensure an especially low FWER by first setting $\alpha$ very low, *and then* getting a very precise estimate of the corresponding threshold. The smallest possible $p$-value we can derive this way is $1/T$, so for very low $p$-values, $T$ must be very large.

### 2.2 Low-rank Matrix completion

Low-rank matrix completion [19] seeks to reconstruct missing entries from a matrix, given only a small fraction of its entries. The problem is ill-posed unless we assume this matrix has a low-rank column space. If so, then a much smaller number of observations, on the order of $r \log(v)$, where $r$ is the column space's rank, and $v$ is its ambient dimension [19] is sufficient to recover both an orthogonal basis for the row space as well as the expansion coefficients for each column, giving the recovery. By placing an $\ell_1$-norm penalty on the eigenvalues of the recovered matrix via the nuclear norm [20, 21] we can ensure that the solution is as low rank as possible. Alternatively, we can specify a rank $r$ ahead of time, and estimate an orthogonal basis of that rank by following a gradient along the Grassmannian manifold [22, 17]. Denoting the set of randomly subsampled entries as $\Omega$, the matrix completion problem is given as,

$$\min_{\tilde{\mathbf{P}}} \|\mathbf{P}_\Omega - \tilde{\mathbf{P}}_\Omega\|_F^2 \qquad \text{s.t. } \tilde{\mathbf{P}} = \mathbf{U}\mathbf{W}; \ \mathbf{U} \text{ is orthogonal} \tag{1}$$

where $\mathbf{U} \in \mathbb{R}^{v \times r}$ is the low-rank basis of $\mathbf{P}$, $\Omega$ gives the measured entries, and $\mathbf{W}$ is the set of expansion coefficients which reconstructs $\tilde{\mathbf{P}}$ in $\mathbf{U}$. Two recent methods operate in an online setting, i.e., where rows of $\mathbf{P}$ arrive one at a time, and both $\mathbf{U}$ and $\mathbf{W}$ are updated accordingly [22, 17].

### 2.3 Low rank plus a long tail

Real-world data often have a dominant low-rank component. While the data may not be *exactly* characterized by a low-rank basis, the residual will not significantly alter the eigen-spectrum of the sample covariance in such cases. Having strong correlations is nearly synonymous with having a

skewed eigen-spectrum, because the flatter the eigen-spectrum becomes, the sparser the resulting covariance matrix tends to be (the "uncertainty principle" between low-rank and sparse matrices [23]). This low-rank structure carries through for purely linear statistics (such as sample means). However, non-linearities in the test statistic calculation, e.g., normalizing by pooled variances, will contribute a long tail of eigenvalues, and so we require that this long tail will either decay rapidly, or that it does not overlap with the dominant eigenvalues. For $t$-statistics, the pooled variances are unlikely to change very much from one permutation sample to another (barring outliers) — hence we expect that the spectrum of $\mathbf{P}$ will resemble that of the data covariance, with the addition of a long, exponentially decaying tail. More generally, if the non-linearity does not de-correlate the test statistics too much, it will preserve the low-rank structure.

If this long tail is indeed dominated by the low-rank structure, then its contribution to $\mathbf{P}$ can be modeled as a low variance Gaussian i.i.d. residual. A Central Limit argument appeals to the number of independent eigenfunctions that contribute to this residual, and, the orthogonality of eigenfunctions implies that as more of them meaningfully contribute to each entry in the residual, the more independent those entries become. In other words, if this long tail begins at a low magnitude and decays slowly, then we can treat it as a Gaussian i.i.d. residual; and if it decays rapidly, then the residual will perhaps be less Gaussian, but also more negligible. Thus, our development in the next section makes no direct assumption about these eigenvalues themselves, but rather that the residual corresponds to a low-variance i.i.d. Gaussian random matrix — its contribution to the covariance of test statistics will be Wishart distributed, and from that we can characterize its eigenvalues.

## 2.4 Our Method

It still remains to model the residual numerically. By sub-sampling we can reconstruct the low-rank portion of $\mathbf{P}$ via matrix completion, but in order to obtain the desired sample maximum distribution we must also recover the residual. Exact recovery of the residual is essentially impossible; fortunately, for our purposes we need only need its effect on the distribution of the *maximum per permutation test*. So, we estimate its variance, (its mean is zero by assumption,) and then randomly sample from that distribution to recover the unobserved remainder of the matrix.

A large component in the running time of online subspace tracking algorithms is spent in updating the basis set $\mathbf{U}$; yet, once a good estimate for $\mathbf{U}$ has been found this becomes superfluous. We therefore divide the entire process into two steps: training, and recovery. During the training phase we conduct a small number of fully sampled permutation tests (100 permutations in our experiments). From these permutation tests, we estimate $\mathbf{U}$ using sub-sampled matrix completion methods [22, 17], making multiple passes over the training set (with fixed sub-sampling rate), until convergence. In our evaluations, three passes sufficed. Then, we obtain a distribution of the residual $\mathbf{S}$ over the entire training set. Next is the recovery phase, in which we sub-sample a small fraction of the entries of each successive column $t$, solve for the reconstruction coefficients $\mathbf{W}(\cdot, t)$ in the basis $\mathbf{U}$ by least-squares, and then add random residuals using parameters estimated during training. After that, we proceed exactly as in a normal permutation testing, to recover the statistics.

*Bias-Variance tradeoff.* By using a very sparse subsampling method, there is a bias-variance dilemma in estimating $\mathbf{S}$. That is, if we use the entire matrix $\mathbf{P}$ to estimate $\mathbf{U}$, $\mathbf{W}$ and $\mathbf{S}$, we will obtain reliable estimates of $\mathbf{S}$. But, there is an overfitting problem: the least-squares objective used in fitting $\mathbf{W}(\cdot, t)$ to such a small sample of entries is likely to grossly underestimate the variance of $\mathbf{S}$ compared to where we use the entire matrix; (the sub-sampling problem is not nearly as over-constrained as for the whole matrix). This sampling artifact reduces the apparent variance of $\mathbf{S}$, and induces a bias in the distribution of the sample maximum, because extreme values are found less frequently. This sampling artifact has the effect of 'shifting' the distribution of the sample maximum towards 0. We correct for this bias by estimating the amount of the shift during the training phase, and then shifting the recovered sample max distribution by this estimated amount.

## 3 Analysis

We now discuss two results which show that as long as the variance of the residual is below a certain level, we can recover the distribution of the sample maximum. Recall from (1) that for low-rank matrix completion methods to be applied we must assume that the permutation matrix $\mathbf{P}$ can be

decomposed into a low-rank component plus a high-rank residual matrix $\mathbf{S}$:

$$\mathbf{P} = \mathbf{UW} + \mathbf{S}, \tag{2}$$

where $\mathbf{U}$ is a $v \times r$ orthogonal matrix that spans the $r \ll \min(v, t)$ -dimensional column subspace of $\mathbf{P}$, and $\mathbf{W}$ is the corresponding coefficient matrix. We can then treat the residual $\mathbf{S}$ as a random matrix whose entries are i.i.d. zero-mean Gaussian with variance $\sigma^2$. We arrive at our first result by analyzing how the low-rank portion of $\mathbf{P}$'s singular spectrum interlaces with the contribution coming from the residual by treating $\mathbf{P}$ as a low-rank perturbation of a random matrix. If this low-rank perturbation is sufficient to dominate the eigenvalues of the random matrix, then $\mathbf{P}$ can be recovered with high fidelity at a low sampling rate [22, 17]. Consequently, we can estimate the distribution of the maximum as well, as shown by our second result.

The following development relies on the observation that the eigenvalues of $\mathbf{PP}^T$ are the squared singular values of $\mathbf{P}$. Thus, rather than analyzing the singular value spectrum of $\mathbf{P}$ directly, we can analyze the eigenvalues of $\mathbf{PP}^T$ using a recent result from [24]. This is important because in order to ensure recovery of $\mathbf{P}$, we require that its singular value spectrum will approximately retain the shape of $\mathbf{UW}$'s. More precisely, we require that for some $0 < \delta < 1$,

$$|\tilde{\phi}_i - \phi_i| < \delta\phi_i \qquad i = 1, \dots, r; \qquad \tilde{\phi}_i < \delta\phi_r \qquad i = r+1, \dots, v \tag{3}$$

where $\phi_i$ and $\tilde{\phi}_i$ are the singular values of $\mathbf{UW}$ and $\mathbf{P}$ respectively. (Recall that in this analysis $\mathbf{P}$ is the perturbation of $\mathbf{UW}$.) Thm. 3.1 relates the rate at which eigenvalues are perturbed, $\delta$, to the parameterization of $\mathbf{S}$ in terms of $\sigma^2$. The theorem's principal assumption also relates $\sigma^2$ inversely with the number of columns of $\mathbf{P}$, which is just the number of trials $t$. Note however that the process may be split up between several matrices $\mathbf{P}_i$, and the results can then be combined. For purposes of applying this result in practice we may then choose a number of columns $t$ which gives the best bound. Theorem 3.1 also assumes that the number of trials $t$ is greater than the number of voxels $v$, which is a difficult regime to explore empirically. Thus, our numerical evaluations cover the case where $t < v$, while Thm 3.1 covers the case where $t$ is larger.

From the definition of $\mathbf{P}$ in (2), we have,

$$\mathbf{PP}^T = \mathbf{UWW}^T\mathbf{U}^T + \mathbf{SS}^T + \mathbf{UWS}^T + \mathbf{SW}^T\mathbf{U}^T. \tag{4}$$

We first analyze the change in eigenvalue structure of $\mathbf{SS}^T$ when perturbed by $\mathbf{UWW}^T\mathbf{U}^T$, (which has $r$ non-zero eigenvalues). The influence of the cross-terms ($\mathbf{UWS}^T$ and $\mathbf{SW}^T\mathbf{U}^T$) is addressed later. Thus, we have the following theorem.

**Theorem 3.1.** *Denote that $r$ non-zero eigenvalues of $\mathbf{Q} = \mathbf{UWW}^T\mathbf{U}^T \in \mathbb{R}^{v \times v}$ by $\lambda_1 \geq \lambda_2 \geq , \dots, \lambda_r > 0$; and let $\mathbf{S}$ be a $v \times t$ random matrix such that $\mathbf{S}_{i,j} \sim \mathcal{N}(0, \sigma^2)$, with unknown $\sigma^2$. As $v, t \to \infty$ such that $\frac{v}{t} \ll 1$, the eigenvalues $\tilde{\lambda}_i$ of the perturbed matrix $\mathbf{Q} + \mathbf{SS}^T$ will satisfy*

$$|\tilde{\lambda}_i - \lambda_i| < \delta\lambda_i \qquad i = 1, \dots, r; \qquad \tilde{\lambda}_i < \delta\lambda_r \qquad i = r+1, \dots, v \tag{$\star$}$$

*for some $0 < \delta < 1$, whenever $\sigma^2 < \frac{\delta\lambda_r}{t}$*

*Proof. (Sketch)* The proof proceeds by constructing the asymptotic eigenvalues $\tilde{\lambda}_i$ (for $i = 1, \dots, v$), and later bounding them to satisfy ($\star$). The construction of $\tilde{\lambda}_i$ is based on Theorem 2.1 from [24]. Firstly, an asymptotic spectral measure $\mu$ of $\frac{1}{t}\mathbf{SS}^T$ is calculated, followed by its Cauchy transform $G_\mu(z)$. Using $G_\mu(z)$ and its functional inverse $G_\mu^{-1}(\theta)$, we get $\tilde{\lambda}_i$ in terms of $\lambda_i$, $\sigma^2$, $v$ and $t$. Finally, the constraints in ($\star$) are applied to $\tilde{\lambda}_i$ to upper bound $\sigma^2$. The supplement includes the proof. □

Note that the missing cross-terms would not change the result of Theorem 3.1 drastically, because $\mathbf{UW}$ has $r$ non-zero singular values and hence $\mathbf{UWS}^T$ is a low-rank projection of a low-variance random matrix, and this will clearly be dominated by either of the other terms. Having justified the model in (2), the following thorem shows that the empirical distribution of the maximum Null statistic approximates the true distribution.

**Theorem 3.2.** *Let $m_t = \max_i \mathbf{P}_{i,t}$ be the maximum observed test statistic at permutation trial $t$, and similarly let $\hat{m}_t = \max_i \hat{\mathbf{P}}_{i,t}$ be the maximum reconstructed test statistic. Further, let the maximum reconstruction error be $\epsilon$, such that $|\mathbf{P}_{i,t} - \hat{\mathbf{P}}_{i,t}| \leq \epsilon$. Then, for any real number $k > 0$, we have,*

$$Pr\left[m_t - \hat{m}_t - (b - \hat{b}) > k\epsilon\right] < \frac{1}{k^2}$$

*where $b$ is the bias term described in Section 2, and $\hat{b}$ is its estimate from the training phase.*

The result is an application of Chebyshev's bound. The complete proof is given in the supplement.

## 4 Experimental evaluations

Our experimental evaluations include four separate neuroimaging datasets of Alzheimer's Disease (AD) patients, cognitively healthy age-matched controls (CN), and in some cases Mild Cognitive Impairment (MCI) patients. The first of these is the Alzheimer's Disease Neuroimaging Initiative (ADNI) dataset, a nation-wide multi-site study. ADNI is a landmark study sponsored by the NIH, major pharmaceuticals and others to determine the extent to which multimodal brain imaging can help predict onset, and monitor progression of AD. The others were collected as part of other studies of AD and MCI. We refer to these datasets as Dataset A—D. Their demographic characteristics are as follows: Dataset A: 40 subjects, AD vs. CN, median age : 76; Dataset B: 50 subjects, AD vs. CN, median age : 68; Dataset C: 55 subjects, CN vs. MCI, median age : 65.16; Dataset D: 70 subjects, CN vs. MCI, median age : 66.24.

Our evaluations focus on three main questions: **(i)** Can we recover an acceptable approximation of the maximum statistic Null distribution from an approximation of the permutation test matrix? **(ii)** What degree of computational speedup can we expect at various subsampling rates, and how does this affect the trade-off with approximation error? **(iii)** How sensitive is the estimated $\alpha$-level threshold with respect to the recovered Null distribution? In all our experiments, the rank estimate for subspace tracking (to construct the low–rank basis $\mathbf{U}$) was taken as the number of subjects.

### 4.1 Can we recover the Maximum Null?

Our experiments suggest that our model can recover the maximum Null. We use Kullback–Leibler (KL) divergence and Bhattacharya Distance (BD) to compare the estimated maximum Null from our model to the true one. We also construct a "Naive–Null", where the subsampled statistics are pooled and the Null distribution is constructed with no further processing (i.e., completion). Using this as a baseline, Fig. 1 shows the KL and BD values obtained from three datasets, at 20 different subsampling rates (ranging from $0.1\%$ to $10\%$). Note that our model involves a training module where the approximate 'bias' of residuals is estimated. This estimation is prone to noise (for example, number of training frames). Hence Fig. 1 also shows the error bars pertaining to 5 realizations on the 20 sampling rates. The first observation from Fig. 1 is that both KL and BD measures of the recovered Null to the true distribution are $< e^{-5}$ for sampling rates more than $0.4\%$. This

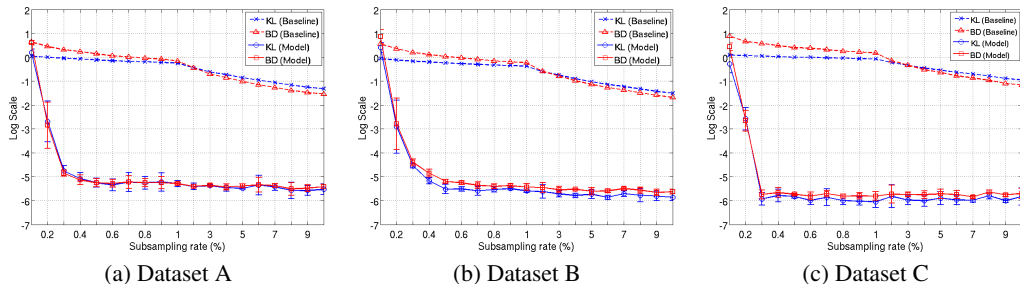

|            (a) Dataset A            |            (b) Dataset B            |            (c) Dataset C            |

Figure 1: KL (blue) and BD (red) measures between the true max Null distribution (given by the full matrix **P**) and that recovered by our method (thick lines), along with the baseline naive subsampling method (dotted lines). Results for Datasets A, B, C are shown here. Plot for Dataset D is in the extended version of the paper.

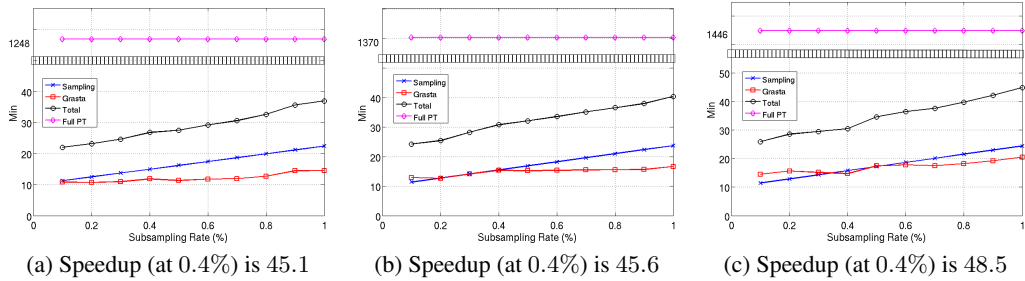

(a) Speedup (at $0.4\%$) is $45.1$    (b) Speedup (at $0.4\%$) is $45.6$    (c) Speedup (at $0.4\%$) is $48.5$

Figure 3: Computation time (in minutes) of our model compared to that of computing the entire matrix $\mathbf{P}$. Results are for the same three datasets as in Fig. 1. Please find the plot for Dataset D in the extended version of the paper. The horizontal line (magenta) shows the time taken for computing the full matrix $\mathbf{P}$. The other three curves include : subsampling (blue), GRASTA recovery (red) and total time taken by our model (black). Plots correspond to the low sampling regime ($< 1\%$) and note the jump in y–axis (black boxes). For reference, the speedup factor at $0.4\%$ sampling rate is reported at the bottom of each plot.

suggests that our model recovers both the shape (low BD) and position (low KL) of the null to high accuracy at extremely low sub-sampling. We also see that above a certain minimum subsampling rate ($\sim 0.3\%$), the KL and BD do not change drastically as the rate is increased. This is expected from the theory on matrix completion where after observing a minimum number of data samples, adding in new samples does not substantially increase information content. Further, the error bars (although very small in magnitude) of both KL and BD show that the recovery is noisy. We believe this is due to the approximate estimate of bias from training module.

## 4.2 What is the computational speedup?

Our experiments suggest that the speedup is substantial. Figs. 3 and 2 compare the time taken to perform the complete permutation testing to that of our model. The three plots in Fig. 3 correspond to the datasets used in Fig. 1, in that order. Each plot contains 4 curves and represent the time taken by our model, the corresponding sampling and GRASTA [17] recovery (plus training) times and the total time to construct the entire matrix $\mathbf{P}$ (horizontal line). And Fig. 2 shows the scatter plot of computational speedup vs. KL divergence (over 3 repeated set of experiments on all the datasets and sampling rates). Our model achieved at least 30 times decrease in computation time in the low sampling regime ($< 1\%$). Around $0.5\% - 0.6\%$ sub-sampling (where the KL and BD are already $< e^{-5}$), the computation speed-up factor averaged over all datasets was $45\times$. This shows that our model achieved good accuracy (low KL and BD) together with high computational speed up

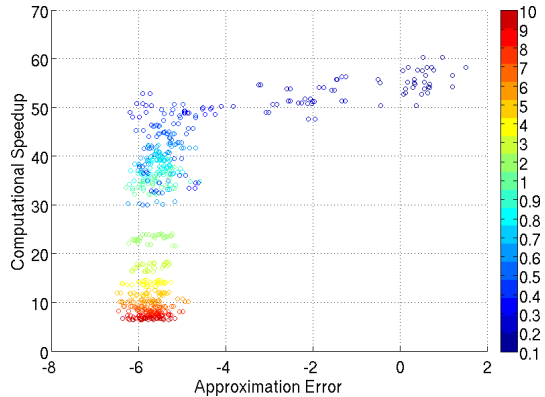

Figure 2: Scatter plot of computational speedup vs. KL. The plot corresponds to the 20 different samplings on all 4 datasets (for 5 repeated set of experiments) and the colormap is from $0.1\%$ to $10\%$ sampling rate. The x–axis is in log scale.

in tandem, especially, for $0.4\% - 0.7\%$ sampling rates. However note from Fig. 2 that there is a trade–off between the speedup factor and approximation error (KL or BD). Overall the highest computational speedup factor achieved at a recovery level of $e^{-5}$ on KL and BD is around 50x (and this occured around $0.4\% - 0.5\%$ sampling rate, refer to Fig. 2). It was observed that a speedup factor of upto $55\times$ was obtained for Datasets C and D at $0.3\%$ subsampling, where the KL and BD were as low as $e^{-5.5}$ (refer to Fig. 1 and the extended version of the paper).

## 4.3 How stable is the estimated $\alpha$-threshold (clinical significance)?

Our experiments suggest that the threshold is stable. Fig. 4 and Table 1 summarize the clinical significance of our model. Fig. 4 show the error in estimating the true max thresh-

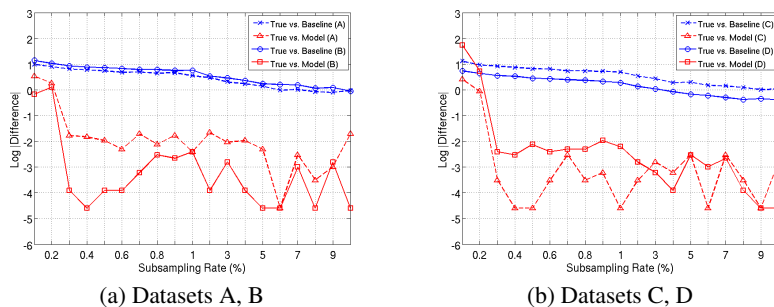

(a) Datasets A, B         (b) Datasets C, D

Figure 4: Error of estimated $t$ statistic thresholds (red) for the 20 different subsampling rates on the four Datasets. The confidence level is $1 - \alpha = 0.95$. The y-axis is in log–scale. For reference, the thresholds given by baseline model (blue) are included. Note that each plot corresponds to two datasets.

old, at $1 - \alpha = 0.95$ level of confidence. The x–axis corresponds to the 20 different sampling rates used and y–axis shows the absolute difference of thresholds in log scale. Observe that for sampling rates higher than $3\%$, the mean and maximum differences was $0.04$ and $0.18$. Note that the binning resolution of max.statistic used for constructing the Null was $0.01$. These results show that not only the global shape of the maximum Null distribution is estimated to high accuracy (see Section 4.1) but also the shape and area in the tail. To support this observation, we show the absolute differences of the estimated thresholds on all the datasets at 4 different $\alpha$ levels in Table 1. The errors for $1 - \alpha = 0.95, 0.99$ are at most $0.16$. The increase in error for $1 - \alpha > 0.995$ is a sampling artifact and is expected. Note that in

| Data name | Sampling rate | $1 - \alpha$ level | | | |
|---|---|---|---|---|---|
| | | 0.95 | 0.99 | 0.995 | 0.999 |
| $A$ | 0.3% | 0.16 | 0.11 | 0.14 | 0.07 |
| | 0.5% | 0.13 | 0.08 | 0.10 | 0.03 |
| $B$ | 0.3% | 0.02 | 0.05 | 0.03 | 0.13 |
| | 0.5% | 0.02 | 0.07 | 0.08 | 0.04 |
| $C$ | 0.3% | 0.04 | 0.13 | 0.21 | 0.20 |
| | 0.5% | 0.01 | 0.07 | 0.07 | 0.05 |
| $D$ | 0.3% | 0.08 | 0.10 | 0.27 | 0.31 |
| | 0.5% | 0.12 | 0.13 | 0.25 | 0.22 |

Table 1: Errors of estimated $t$ statistic thresholds on all datasets at two different subsampling rates.

a few cases, the error at $0.5\%$ is slightly higher than that at $0.3\%$ suggesting that the recovery is noisy (see Sec. 4.1 and the errorbars of Fig. 1). Overall the estimated $\alpha$-thresholds are both faithful and stable.

## 5 Conclusions and future directions

In this paper, we have proposed a novel method of efficiently approximating the permutation testing matrix by first estimating the major singular vectors, then filling in the missing values via matrix completion, and finally estimating the distribution of residual values. Experiments on four different neuroimaging datasets show that we can recover the distribution of the maximum Null statistic to a high degree of accuracy, while maintaining a computational speedup factor of roughly $50\times$. While our focus has been on neuroimaging problems, we note that multiple testing and False Discovery Rate (FDR) correction are important issues in genomic and RNA analyses, and our contribution may offer enhanced leverage to existing methodologies which use permutation testing in these settings[6].

**Acknowledgments:** We thank Robert Nowak, Grace Wahba, Moo K. Chung and the anonymous reviewers for their helpful comments, and Jia Xu for helping with a preliminary implementation of the model. This work was supported in part by NIH R01 AG040396; NSF CAREER grant 1252725; NSF RI 1116584; Wisconsin Partnership Fund; UW ADRC P50 AG033514; UW ICTR 1UL1RR025011 and a Veterans Administration Merit Review Grant I01CX000165. Hinrichs is supported by a CIBM post-doctoral fellowship via NLM grant 2T15LM007359. The contents do not represent views of the Dept. of Veterans Affairs or the United States Government.

## Footnotes

*Hinrichs and Ithapu are joint first authors and contributed equally to this work.

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
