[Supplementary Material]

# Supplement

## Proof of Theorem 3.1

*Denote that $r$ non-zero eigenvalues of $\mathbf{Q} = \mathbf{U}\mathbf{W}\mathbf{W}^T\mathbf{U}^T \in \mathbb{R}^{v \times v}$ by $\lambda_1 \geq \lambda_2 \geq, \ldots, \lambda_r > 0$; and let $\mathbf{S}$ be a $v \times t$ random matrix such that $S_{i,j} \sim \mathcal{N}(0, \sigma^2)$, with unknown $\sigma^2$. As $v, t \to \infty$ such that $\frac{v}{t} \ll 1$, the eigenvalues $\tilde{\lambda}_i$ of the perturbed matrix $\mathbf{Q} + \mathbf{S}\mathbf{S}^T$ will satisfy*

$$|\tilde{\lambda}_i - \lambda_i| < \delta\lambda_i \qquad i = 1, \ldots, r; \qquad \tilde{\lambda}_i < \delta\lambda_r \qquad i = r+1, \ldots, v \qquad (\star)$$

*for some $0 < \delta < 1$, whenever $\sigma^2 < \frac{\delta\lambda_r}{t}$*

*Proof.* The first half of the proof emulates Theorem 2.1 from [1]. Consider the matrix $\mathbf{X} = \sqrt{t}\mathbf{S}$. By the structure of $\mathbf{S}$, each entry of $\mathbf{X}$ is i.i.d. Gaussian with zero–mean and variance $\sigma^2 t$. Let $\mathbf{Y} = \frac{1}{t}\mathbf{X}\mathbf{X}^T$ and denote its ordered eigenvalues as $\gamma_i, i = 1, \ldots, v$ (large to small). Consider the random spectral measure

$$\mu_v(A) = \tfrac{1}{v}\#\{\gamma_i \in A\}, \qquad A \subset \mathbb{R}$$

The Marchenko–Pastur law [2] states that as $v, t \to \infty$ such that $\frac{v}{t} \leq 1$, the random measure $\mu_v \to \mu$, where $d\mu$ is given by

$$d\mu(a) = \tfrac{1}{2\pi\sigma^2 t\gamma a}\sqrt{(\gamma_+ - a)(a - \gamma_-)}\mathbf{1}_{[\gamma_-, \gamma_+]}da$$

where $\gamma = \frac{v}{t}$. Here $\mathbf{1}_{[\gamma_-, \gamma_+]}$ is an indicator function that is non–zero on $[\gamma_-, \gamma_+]$. $\gamma_\pm = \sigma^2 t(1 \pm \sqrt{\gamma})^2$ are the extreme points of the support of $\mu$. It is well known that the extreme eigenvalues converge almost surely to $\gamma_\pm$ [3]. Since $v, t \to \infty$ and $\gamma = \frac{v}{t} \ll 1$, the length of $[\gamma_-, \gamma_+]$ is much smaller than the values in it. Hence we have,

$$\gamma_\pm \sim \sigma^2 t(1 \pm 2\sqrt{\gamma}) \quad ; \quad \sqrt{(\gamma_+ - a)(a - \gamma_-)} \ll a$$

and the new $d\mu(a)$ is given by

$$d\mu(a) = \frac{\sqrt{(\sigma^2 t(1 + 2\sqrt{\gamma}) - a)(a - \sigma^2 t(1 - 2\sqrt{\gamma}))}}{2\pi\gamma\sigma^4 t^2}\mathbf{1}_{[\sigma^2 t(1 - 2\sqrt{\gamma}), \sigma^2 t(1 + 2\sqrt{\gamma})]}da$$

$$= \frac{1}{2\pi\gamma\sigma^4 t^2}\sqrt{4\gamma\sigma^4 t^2 - (a - \sigma^2 t)^2}\mathbf{1}_{[\sigma^2 t(1 - 2\sqrt{\gamma}), \sigma^2 t(1 + 2\sqrt{\gamma})]}da$$

The form we have derived for $d\mu(a)$ shares some similarities with $d\mu_X(x)$ in Section 3.1 of [1]. The analysis in [1] takes into account the phase transition of extreme eigen values. This is done by imitating a time–frequency type analysis on compact support of extreme spectral measure i.e. using Cauchy transform. For our case, the Cauchy transform of $\mu(a)$ is

$$G_\mu(z) = \frac{1}{2\gamma\sigma^4 t^2}\left(z - \sigma^2 t - sgn(z)\sqrt{(z - \sigma^2 t)^2 - 4\gamma\sigma^4 t^2}\right)$$

$$\text{for} \quad z \in (\infty, \sigma^2 t(1 - 2\sqrt{\gamma})) \cup (\sigma^2 t(1 + 2\sqrt{\gamma}), \infty)$$

Since we are interested in the asymptotic eigen values (and $\gamma \ll 1$), $G_\mu(\gamma_\pm)$ and the functional inverse $G_\mu^{-1}(\theta)$ are

$$G_\mu(\gamma_+) = \frac{1}{\sigma^2 t \sqrt{(\gamma)}} \quad ; \quad G_\mu(\gamma_-) = -\frac{1}{\sigma^2 t \sqrt{(\gamma)}} \quad ; \quad G_\mu^{-1}(\theta) = \sigma^2 t + \frac{1}{\theta} + \gamma \sigma^4 t^2 \theta$$

Hence, the asymptotic behavior of the eigen values of perturbed matrix $\mathbf{Q} + \mathbf{S}\mathbf{S}^T$ is (observing that $\mathbf{S}\mathbf{S}^T = \mathbf{Y}$ and $\mathbf{Q}$ has $r$ non–zero positive eigen values)

$$\tilde{\lambda}_i(i = 1, \ldots, r) \approx \begin{cases} \lambda_i + \sigma^2 t + \frac{\gamma \sigma^4 t^2}{\lambda_i} & \text{for} \quad \lambda_i > \gamma \sigma^2 t \\ \gamma \sigma^2 t & \text{else} \end{cases} \tag{$*$}$$

$$\tilde{\lambda}_i(i = r+1, \ldots, v) \approx \sigma^2 t(1 - 2\sqrt{(\gamma)})$$

With $\tilde{\lambda}_i, i = 1, \ldots, v$ in hand, we now bound the unknown variance $\sigma^2$ such that $(\star)$ is satisfied. We only have two cases to consider,

$$(1) \quad \lambda_i > \gamma \sigma^2 t \, , i = 1, \ldots, r \qquad (2) \quad \lambda_i \leq \gamma \sigma^2 t \, , i = k, \ldots, r \text{ (for some } k \geq 1)$$

We constrain the unknown $\sigma^2$ such that case (2) does not arise. Substituting for $\tilde{\lambda}_i$'s from $(*)$ in $(\star)$, we get,

$$\sigma^2 t + \frac{\gamma \sigma^4 t^2}{\lambda_i} < \delta \lambda_i \quad ; \quad \lambda_i > \gamma \sigma^2 t \quad ; \quad \sigma^2 t(1 - 2\sqrt{(\gamma)}) < \delta \lambda_r$$

These inequalities will hold when $\sigma^2 < \frac{\delta \lambda_r}{t}$ (since $\gamma = \ll 1, \delta < 1$ and $\lambda_1 \geq \lambda_2 \geq, \ldots, \lambda_r$).

$\square$

## Proof of Theorem 3.2

*Let $m_t = \max_i \mathbf{P}_{i,t}$ be the maximum observed test statistic at permutation trial $t$, and similarly let $\hat{m}_t = \max_i \hat{\mathbf{P}}_{i,t}$ be the maximum reconstructed test statistic. Further, let the maximum reconstruction error be $\epsilon$, such that $|\mathbf{P}_{i,t} - \hat{\mathbf{P}}_{i,t}| \leq \epsilon$. Then, for any real number $k > 0$, we have,*

$$Pr\left[ m_t - \hat{m}_t - (b - \hat{b}) > k\epsilon \right] < \frac{1}{k^2}$$

*where $b$ is the bias term described in Section 2, and $\hat{b}$ is its estimate from the training phase.*

*Proof.* Recall that there is a bias term in estimating the distribution of the maximum which must be corrected for this is because $\text{var}(\hat{S})$ underestimates $\text{var}(S)$ due to the bias/variance tradeoff. Let $b$ be this difference:

$$b = \mathbb{E}_t\left[ \max_i \mathbf{P}_{i,t} \right] - \mathbb{E}_t\left[ \max_i \hat{\mathbf{P}}_{i,t} \right].$$

Further, recall that we estimate $b$ by taking the difference of mean sample maxima between observed and reconstructed test statistics over the training set, giving $\hat{b}$, which is an unbiased estimator of $b$ — it is unbiased because a difference in sample means is an unbiased estimator of the difference of two expectations.

Let $\delta_t = m_t - \hat{m}_t$. To show the result we must derive a concentration bound on $\delta_t$, which we will do by applying Chebyshev's inequality. In order to do so, we require an expression for the mean and variance of $\delta_t$. First, we derive an expression for the mean. Taking the expectation over $t$ of $m_t - \hat{m}_t$ we have,

$$\mathbb{E}_t[m_t - \hat{m}_t] = \mathbb{E}_t\left[ \max_i \mathbf{P}_{i,t} - \max_i \hat{\mathbf{P}}_{i,t} - \hat{b} \right]$$
$$= \mathbb{E}_t\left[ \max_i \mathbf{P}_{i,t} \right] - \mathbb{E}_t\left[ \max_i \hat{\mathbf{P}}_{i,t} \right] - \hat{b}$$
$$= b - \hat{b}$$

where the second equality follows from the linearity of expectation.

Next, we require an expression for the variance of $\delta_t$. Let $i$ be the index at which the maximum observed test statistic occurs for permutation trial $t$, and likewise let $j$ be the index at which the maximum reconstructed test statistic occurs. Thus we have,

$$\mathbf{P}_{i,t} \leq \hat{\mathbf{P}}_{i,t} + \epsilon \ \leq \hat{\mathbf{P}}_{j,t} + \epsilon$$
$$\mathbf{P}_{i,t} \geq \mathbf{P}_{j,t} \qquad \geq \hat{\mathbf{P}}_{j,t} - \epsilon,$$

and so we have that

$$|m_t - \hat{m}_t| < 2\epsilon$$

and so

$$\mathrm{var}(m_t - \hat{m}_t) \leq \epsilon^2.$$

Applying Chebyshev's bound,

$$\mathrm{Pr}\left[m_t - \hat{m}_t - (b - \hat{b}) > k\epsilon\right] < \frac{1}{k^2}$$

which completes the proof.

$\square$

**Fig 1. : All 4 datasets**

(a) Dataset A

(b) Dataset B

(c) Dataset C

(d) Dataset D

Figure 1: KL (blue) and BD (red) measures between the true max. null distribution (given by the full matrix $\mathbf{P}$) and that recovered by our method (thick lines), along with the baseline naive subsampling method (dotted lines). Each plot corresponds to one of the four datasets used in our evaluations. Note that the y–axis is in log scale.

**Fig 2. : All 4 datasets**

(a) Speedup (at 0.4%) - 45.1

(b) Speedup (at 0.4%) - 45.6

(c) Speedup (at 0.4%) - 48.5

(d) Speedup (at 0.4%) - 56.4

Figure 2: Computation time of our model compared to that of computing the entire matrix **P**. Each plot corresponds to one of the four datasets A, B, C and D (in that order). The horizontal line (magenta) shows the time taken for computing the full matrix **P**. The other three curves include : subsampling (blue), GRASTA recovery (red) and total time taken by our model (black). Plots correspond to the low sampling regime ($< 1\%$) and note the jump in y–axis (black boxes). For reference the speedup at $0.4\%$ sampling rate is reported at the bottom of each plot.