[Reviews · NeurIPS 2013]

Submitted by Assigned_Reviewer_2

Permutation testing is gaining increasing interest in several communities, including the bio-informatics community and the neuroimaging community served specifically in the present contribution. Although one can have serious doubts about the use of permutation test in complex data (users go fishing for tests quantities that make their findings significant etc), permutation testing is basically a sounder approach than parametric tests, as it is based on less restrictive distributional assumptions.

The computational complexity is significant for permutation testing so tools for more efficient use of samples are of high interest. In the present work, the tool becomes even more interesting to NIPS as it gives rise to a new application of unsupervised machine learning/matrix completion.


Quality:
The background and state of the art are quite well covered as far as I can tell. The overall motivation of the work is strong.

However, the motivation and the explanation of how permutation tests are used in the specific case of neuroimaging are unnecessarily talkative (second paragraph of Section 1) and could be simplified and made more precise. The general relation to (Fisher’s) exact tests could have been used (this link is missing). I would also have liked to see a discussion of weak and strong control in neuroimaging (this work only considers weak control, i.e., false alarm control when the global null is true).

Unfortunately, the more detailed assumptions of the work could have been more carefully examined in both simulations and real data. E.g. the central assumption of Theorem 3.1: Here normality of residuals is assumed. This assumption -when and how it breaks down - could have been checked in the simulation. How damaging is it to the bound? In any case Theorem 3.1 is only a “guide for intuition” as it relevant in the wrong limit so to speak (# voxels << # permutation samples). Then, the next question: Is Theorem 3.2 relevant for the experiments?. I.e., what was \epsilon? an how does the maximum reconstruction error depend on of t?

The empirical study is quite rich and detailed, and results are promising. However, the relevance of the theoretical work (assumptions) could have been more detailed argued and validated in the experiments. This is more interesting in the machine learning context than the nice specific empirical findings or "impressive" speed up factors.

Clarity: Paper is quite well written, but could have been more precise. At times it becomes very “talkative” and looses precision (in particular Section 1 and Subsection 2.1).

Originality: The central idea of speeding up permutation testing by modeling the (v \times t) matrix P (unfortunately here dubbed the “permutation matrix” please change to e.g. “permutation test matrix") is new to this reviewer. It is clearly related to existing (referenced) approaches for estimating the effective number of tests. These are also based on low rank approximations, but of *data* rather than approximating the test matrix P.

When the core ideas of the present paper are more fully explored significance will likely be high. In particular if good robust software is released, as promised by the authors!. However, at present the work has too many loose ends.
Summary: An original idea with quite significant practical potential. Nice experimental work and results. Theoretical work is weaker and the critical assumptions could have been much more carefully checked. Link to conventional estimates of the effective number of variables could have been desired.

Submitted by Assigned_Reviewer_3

general comments:

i love this paper, and i think it makes a great contribution. in particular, the matrix completion and optimization communities have mostly focused on estimation problems, rather than testing problems. this is a beautiful example of how the novel developments in one field can make substantial contributions to another field. note that i haven't read the supplement and i am not an expert in matrix completion, though i do know some stats and neuroscience.

major comments:

-- the authors comment, "We believe this is due to the approximate estimate of bias from training module." there are several possible sources of error, including model misspecification, approximation error (even given the true model), and sampling error. it will be interesting to investigate when these errors arise, their relative impact, how to mitigate and parse them, etc. some synthetic data analysis would be a most welcome addition; i imagine the full journal length version of this work would include that.

-- t-tests are ubiquitous and great, and obviously problematic. here, perhaps less problematic, as rather than modeling the distribution of the data, the authors are modeling the distribution of the residuals. thus, already, perhaps this test is "robust" in certain ways (to the assumption of gaussian distributed data). yet, robust isn't mentioned in the manuscript. i think this method is robust to certain very simple assumptions, and that point should be highlighted and explored if i am correct.

-- the data sets that were used are perfectly fine. however, no ground truth is available for them. so, to complement a synthetic data analysis, a real data analysis with ground truth would be excellent. my favorite example of such an endeavor is from:
@article{Eklund2012a,
author = {Eklund, Anders and Andersson, Mats and Josephson, Camilla and Johannesson, Magnus and Knutsson, Hans},
journal = {NeuroImage},
month = apr,
number = {3},
pages = {565--578},
pmid = {22507229},
publisher = {Elsevier Inc.},
title = {{Does parametric fMRI analysis with SPM yield valid results?-An empirical study of 1484 rest datasets.}},
url = {http://www.ncbi.nlm.nih.gov/pubmed/22507229},
volume = {61},
year = {2012}
}

using this data, one can check how accurate the permutation tests are, and then how much accuracy degrades via this method.

-- the figures could be improved. the checklist available at: http://buddhabaki.tumblr.com/post/44143989765/top-20-figure-checklist could help. specifically, fonts are too small, graphics are not vector graphics, symbols are not so obviously different, color choices might be hard for color blind people, axis labels and titles are missing or could be improved.


streaming comments:

-- Bonferoni, FWER, and other statistical techniques should be defined prior to their introduction, as many in this community are relatively unfamiliar with testing.

-- permutation testing is not necessarily sampling from the global null, only in the multiple hypothesis testing scenario. probably worth clarifying.

-- "permutation testing ensures"
does it really "ensure" this? i haven't thought carefully about whether it is possible in the multiple testing scenario for permutation tests to not capture some dependencies, but i imagine it does not. for example, 3rd order dependencies, given finite samples, wouldn't be captured typically. this is not to say that it is not unbiased, but rather, unbiased doesn't ensure relevant dependencies are carried through; those that are hard to capture might still be missed. please cite something or soften or explain.

-- " are required"
don't love that either. required for what? high power? yes. unbiased results, not necessarily. please clarify or soften.

-- "Observe that if voxel-wise statistics were completely i.i.d., then regardless of their distribution, we could simply compute the distribution of the sample maximum analytically by exponentiating."
what do you mean exponentiating the univariate CDF? multiplying? please clarify.
also, i think CDF is not defined, only the acronym is given.

-- why is "Null" capital? and "Global" seems non-standard to me, but maybe is standard in some literatures. certainly not in the stats textbooks i'm familiar with, eg 'testing statistical hypotheses' is the standard used at many top programs, and is my default for notations & definitions.

-- "this subspace. [15, 5]."
extra period.

-- "In other words, our algorithm does the same job as permutation testing,
099 but takes anywhere from a few minutes up to a few hours, rather than days or weeks."
but is approximate, and its accuracy will depend on the matrix completion assumptions and algorithms.


-- Theorem 3.1 of which paper? i figured it out eventually, but was confused at first, fyi. also, you fluctuate between Thm and Theorem; i prefer consistency.

-- "In order to preserve anonymity,"
while i appreciate preserving anonymity, referring to the studies by a letter does not anonymize, afaict. anonymizing involves stripping headers, and possibly faces, generating unique random IDs, etc. also, punctuation is a bit weird in the following sentence, with a space before the colon only sometimes.

-- "How sensitive is the estimated α-level threshold with respect to the recovered Null distribution"
how sensitive to what?

-- fig 1 shows the KL and BD divergence/distance between the subsampled and the truth? i guess that is implied, but i don't see that explicitly stated. for example, y-axis just gives scale, not quantity. no titles. and the caption doesn't make it clear to me either.

-- does 'e^{-5}' mean '10^{-5}'? or really 'e'? i guess that means the log scale is natural log scale. maybe those things are safe to assume in this community, though i wasn't sure.

-- "We believe this is due to the approximate estimate of bias from training module."
it would be great to investigate the source of this error. simulations would obviously be appropriate for such an investigation. a synthetic data analysis, where the simulation parameters were chosen to match the realized data to the extent possible would be especially interesting.

-- i think in your tex file, when writing things like Fig. blah, if you instead write 'Fig.\ ' the spacing will look cleaner (i'm just guessing that you are not)

-- i think fig 2 might be more impressive on a log scale? the black box thing is weird to me. i've often seen a line break with some twiddles separating, know what i mean? also, the y-axis is 'time', the units are '(minutes)'. on a log scale, the speed-up factor may be explicit in the image?

-- i don't think 'upto' is a word.

-- "Our experiments suggest that the threshold is stable."
i think this sentence belongs *after* showing it, not before?


QUALITY: great
CLARITY: great
ORIGINALITY & SIGNIFICANCE: great
Summary: bridging between two disparate fields, and making a useful contribution to a third applied domain, is the highest praise i can give to a paper, which this one recieves.

Submitted by Assigned_Reviewer_4

In this study, the authors aim to speed up permutation test by using random matrix completion. To do this, a subset of the permutation matrix is used to calculate matrices and to obtain a low-rank representation of the full permutation matrix. Since only a subset of the full matrix is required, the method is much faster than calculation of the full permutation matrix. The authors provide theoretical analysis of errors of the completed matrix and test statistics derived from the completed matrix, i.e., maximum statistics for FWER. The method was evaluated in its accuracy and computational time using real neuroscience datasets. The result showed that only a small portion of the permutation matrix (< 1%) were enough to achieve acceptable error rate and its computation was 50 times faster. I think this study is practically important because we need to deal with very large and complex dataset not only in neuroscience, but also in bioinformatics, web data, and so on.
Summary: --- Quality
This study is technically sound and the method was well evaluated theoretically and empirically. I hope the following comments improve the quality of the paper:

1) I worried about whether the method also works for other kind of data having different correlation structure. The correlation structure of the neuroimaging data may be the key for the row-rank matrix completion to work well in the experiments. Since the four datasets in this paper are MRI data, the correlation structure might be similar.

2) You might want to move Fig. 2 to Supplement. I think Fig. 3 is informative enough. Although from Fig. 2 we can see the computational time of training and recovery phases separately, the total time is the most important.

3) L384: “Fig. 4 show…” -> “Fig. 4 shows…”.

4) L359: “And Fig. 3…” -> “Fig. 3…”

--- Clarity
The paper is well written in detail. However, the authors might want to consider shorten the introduction. It was difficult for me to know the main idea of this study. I also suggest moving section 2.3 to late. I think the contents in sections 2.2 and 2.4, i.e., row-rank matrix completion and the method, should be described subsequently.

--- Originality
The application of the matrix completion to permutation test is new.

--- Significance
The theoretical justification of the error bound will encourage empirical neuroscientists to use this method. I hope the software implementing the method in this paper will be released soon as the authors mentioned.

Submitted by Assigned_Reviewer_5

This paper presents an approximation to permutation testing that is claimed to be much faster and quite accurate. The technique is based on describing the voxel X permutation matrix as a product of 2 low-rank matrices plus a Gaussian random matrix. The experimental section seems sufficiently detailed to be compelling. What makes the paper difficult for me to comprehend better is the overly verbose first part, which comes at the expense of a detailed technical exposition of the proposed method. The first half of the paper consists of dense text and includes 2 rather simple equations. Then come two theorems that are rather opaque, and to me don't clearly demonstrate why and when the proposed method is expected to work. It seems that for this conference the reverse is preferable -- a shorter verbal introduction covering motivation, basic idea, and related work, and an expanded mathematical section presenting and demonstrating the theoretical ideas. However, the technique seems novel, and if the speedup turns out to be robust, this could be a significant contribution.

Summary: This is an interesting and potentially significant technique that accelerates permutation testing by approximating the voxel X permutation matrix = low-rank + Gaussian residual. This paper can strongly benefit from a rewrite.
Author Feedback

Author rebuttal: The authors appreciate the reviewers comment about the Introduction (Section 1, 2.1) being verbose. This was an unintended consequence of covering the relevant statistical and neuroimaging background in sufficient detail before introducing the main formulation. Based on the reviewer’s comments, these sections will definitely be shortened and made more precise in the final revision.

======
Reviewer 2:

Q) Gaussian Residuals assumption?
A) The approximation of residuals as Gaussian was _not_ a simplifying assumption for analysis. It was empirically observed on all datasets used in experiments (both by plotting the empirical distributions and also from the eigenvalue structure). We acknowledge the importance of clarifying this issue; following the suggestion, all corresponding plots will appear on the project webpage (alongside implementation) and an extended version of the paper.

======

Q) Setup, assumptions and relevance of Theorems 3.1, 3.2?
A) Theorem 3.1 provides the theoretical basis of our model. Theorem 3.2 bounds the error in recovering the maximum statistic Null distribution.
In the permutation testing setting, the number of tests (t) required for a reliable estimate of Null distribution are tens of thousands and even millions [ref 3,14]. Specifically, in Neuroimaging the number of comparisons (v) (i.e. voxels for images) vary from tens to hundreds of thousands. The case where v >> t is not a valid setup for randomization in this situation because we will not cover the *number of effective independent tests*. When constructing our model we are interested in two specific scenarios:
1) v << t : This is exactly the case considered in Theorem 3.1. Observe that in our experiments the data dimensionality is > 200000. So, to experimentally *verify* this case, we need to run more than a million (and possibly more) tests for ground truth, which is very expensive.
2) v < t or (v ~ t) : This is the more realistic case, and corresponds to our evaluations. The reconstructed P is _extremely_ close to the true P, with a speedup of approximately 50.
This implies that our model works well, for any ‘meaningful’ choice of v and t.

Epsilon in Theorem 3.2 is the maximum difference of the reconstructed test statistic to that of the true one. Theorem 3.1 helps understand the behavior of the model, however, as noted in the paper, there is a bias in recovering the maximum of a given test. A training module estimates this bias. Hence the probability of error in constructing the maximum statistic null distribution is obtained by bounding the probability of the reconstruction difference (i.e., epsilon) concurrently with the bias difference using Chebychev’s inequality.

======

Q) Weak vs. Strong Control:
A) This is an excellent observation. From the perspective of statistical tests in the intended application area, weak control of family wise error is very heavily used which is the setting considered here. We appreciate the reviewer’s suggestion of replicating the model for strong control.

======

Reviewer 3:

Q) Synthetic data and Ground Truth analysis?
A) As noted by the reviewer, analysis on synthetic data will be useful to investigate. Yes, we indeed ran experiments on many toy datasets, but suspected that it may not be interesting to the reader, since multiple large real datasets were included. We take the reviewer’s comment seriously and are happy to provide these in the extended version of the paper.
We thank the reviewer for pointing out the excellent reference Elkund et. al.

======

Q) Robustness and efficiency?
A) As noted by the reviewer, our method does not assume any parametric distribution on the data. However, the residuals are modeled as Gaussian (and the empirical analysis corroborates this design choice). This is particularly interesting since our model can be adapted to any statistic (F, Chi-squared) and any quantile computation (maximum, 99%, 95%), whenever the model assumptions are met.
Given that the results so far are quite promising, we do intend to continue work on fully understanding the estimator’s statistical efficiency and robustness.

======

Reviewer 4:

Q) Relevance of the method to other datasets (apart from neuroimaging)?
A) Our method can be adapted to permutation testing tasks in several application domains, as noted in the other comments.
From private communication, we know of at least two other problems where our model is directly applicable: multiple testing in gene expression analysis, and more broadly in genome wide association studies [ref 5,15]. Detailed experiments assessing these ideas are in the planning stages.

======

We thank Reviewer 5 for the comments. Note that detailed proofs of the two theorems are given in the supplement. We will shorten the initial sections which turned out to be lengthier.